

# A pipeline for assembling low copy nuclear markers from plant genome skimming data for phylogenetic use

Marcelo Reginato

Departamento de Botânica, Instituto de Biociências, Universidade Federal do Rio Grande do Sul, Porto Alegre, Rio Grande do Sul, Brazil

## ABSTRACT

**Background:** Genome skimming is a popular method in plant phylogenomics that do not include a biased enrichment step, relying on random shallow sequencing of total genomic DNA. From these data the plastome is usually readily assembled and constitutes the bulk of phylogenetic information generated in these studies. Despite a few attempts to use genome skims to recover low copy nuclear loci for direct phylogenetic use, such endeavor remains neglected. Causes might include the trade-off between libraries with few reads and species with large genomes (*i.e.*, missing data caused by low coverage), but also might relate to the lack of pipelines for data assembling.

**Methods:** A pipeline and its companion R package designed to automate the recovery of low copy nuclear markers from genome skimming libraries are presented. Additionally, a series of analyses aiming to evaluate the impact of key assembling parameters, reference selection and missing data are presented.

**Results:** A substantial amount of putative low copy nuclear loci was assembled and proved useful to base phylogenetic inference across the libraries tested (4 to 11 times more data than previously assembled plastomes from the same libraries).

**Discussion:** Critical aspects of assembling low copy nuclear markers from genome skims include the minimum coverage and depth of a sequence to be used. More stringent values of these parameters reduces the amount of assembled data and increases the relative amount of missing data, which can compromise phylogenetic inference, in turn relaxing the same parameters might increase sequence error. These issues are discussed in the text, and parameter tuning through multiple comparisons tracking their effects on support and congruence is highly recommended when using this pipeline. The skimmingLoci pipeline (https://github.com/mreginato/skimmingLoci) might stimulate the use of genome skims to recover nuclear loci for direct phylogenetic use, increasing the power of genome skimming data to resolve phylogenetic relationships, while reducing the amount of sequenced DNA that is commonly wasted.

Corresponding author
Marcelo Reginato,
reginatobio@yahoo.com.br

## INTRODUCTION

High-throughput sequencing technologies (HTS) have revolutionized the field of phylogenetics, evolutionary biology, systematics and related areas due to the much higher amount of DNA sequences they can provide to base inferences on relationships among lineages. Phylogenetic data require a constant trade-off between amount of DNA sequenced (total base pairs and number of loci) and breadth (number of taxa; *Dodsworth et al., 2019*). For plant phylogenies, HTS are usually associated with methods to reduce genomic complexity prior to sequencing, given the huge variation of genome sizes, difficulties in genome assembly, and the cost per high-quality genome sequence (*Dodsworth et al., 2019*). Popular strategies to reduce genome complexity include RAD-seq (*Eaton et al., 2017*), RNA-seq (*One Thousand Plant Transcriptomes Initiative, 2019*), target enrichment (*Johnson et al., 2019*) and HYB-seq (*Weitemier et al., 2014*). Genome skimming is an early and still popular approach in plant phylogenomics that do not include a genomic reduction step (*Straub et al., 2012*; *Dodsworth et al., 2019*).

Genome skimming relies on random shallow sequencing of total genomic DNA (gDNA) that results in reliable deep sequencing of the high-copy fraction of the genome: plastome (cpDNA), mitogenome (mtDNA), and repetitive elements (*Straub et al., 2012*). Despite its simplicity, the method became popular in systematics related studies because did not require previous genomic knowledge of the interest group, has a lower cost, and its assembled data constitute an expanded set of molecular markers historically used to build Sanger-based plant phylogenies. The plastome is usually readily assembled from genome skims and constitutes the bulk of phylogenetic information generated in these studies. The mitochondrial genome is less utilized in plant systematics, due to the highly conserved nature of its coding loci, coupled with highly divergent noncoding regions and ubiquitous rearrangements (*Straub et al., 2012*), but a subset of genes can usually be recovered and used (*Henriquez et al., 2014*; *Li et al., 2019*). Among the repetitive nuclear element in genome skims, the ribosomal DNA (rDNA) is also readily assembled and constitutes the major source of nuclear information explored (*Weitemier et al., 2014*; *Fonseca & Lohmann, 2020*). Quantification of other repetitive elements (*e.g.*, transposable elements) can be used to build phylogenies with specific methodology (*Dodsworth et al., 2015*), but have been seldom employed.

Sanger-base plant phylogenies across distinct taxonomic ranks have traditionally been based on plastid and ribosomal markers (*Zimmer & Wen, 2013*; *Davis, Xi & Mathews, 2014*). Thus, one advantage of genome skimming is that the output can be used as backbone data and integrated with the huge amount of Sanger-based data available for an expanded taxonomic breadth. On the other hand, a major drawback is that both cpDNA and rDNA have known issues related to phylogenetic inference, especially when used as the only source of information. Plastids are usually maternally inherited and therefore not comprehensive in tracking the relationships in many plant lineages that include cases of speciation involving hybridization and polyploidy (*Zimmer & Wen, 2013*). The same issue might apply for rDNA, due to different copies being homogenized by concerted evolution. More importantly, the abundant gene tree data available now confirmed theoretical

expectations of high amount of gene tree discordance and the need of using wider sampling of unlinked loci for decisive species tree inference (*Degnan & Rosenberg, 2009*).

Genome skimming has been suggested to provide limited recovery of low copy orthologous nuclear regions for sequence alignment (*Dodsworth et al., 2019*). Its main use has been restricted to characterize conserved nuclear loci for primer or probe design for candidate low copy nuclear markers (*Straub et al., 2012*; *Reginato & Michelangeli, 2016*). Despite some attempts to use low copy nuclear markers from genome skims to base phylogenetic inference (*Besnard et al., 2014*; *Besnard et al., 2018*; *Olofsson et al., 2019*; *Vargas et al., 2019*; *Liu et al., 2021*; *Loiseau et al., 2021*; *Meng et al., 2021*; *Cai, Zhang & Davis, 2022*), this avenue is still neglected. Two major factors might have hampered the use of genome skims to generate low copy nuclear data: lack of genomic information for the group of interest and shallow sequencing. Despite the few low copy loci used in Sanger-based phylogenies (*Zhang et al., 2012*), until very recently most non-model organisms completely lacked information on nuclear genes. Massive efforts to generate taxonomically comprehensive transcriptome data (*e.g.*, onekp.com), full genomes (*Chen et al., 2019*), and associated bioinformatics tools have allowed lineage-specific low copy nuclear markers identification and reference design across non-model angiosperms (*Duarte et al., 2010*; *Chamala et al., 2015*; *Johnson et al., 2016*, *2019*). Thus, these same data and tools can now be used to build references for low copy nuclear loci fishing in genome skims. The second challenge relates to the putative insufficient depth of low copy nuclear markers in genome skims. The concept of sequencing depth (*i.e.*, number of times each base is sequenced) is central to the utilization of NGS data (*Straub et al., 2012*). In genome skims, the sequencing depth of the plastid and mitochondrial genomes will reflect their proportion in sequences obtained from total genomic DNA. Therefore, they will show a relatively deeper sequencing than parts of the genome that are present in single copy (*Straub et al., 2012*). However, the relationship between total amount of DNA sequenced and genome size is highly variable across libraries and lineages, within and across different studies. Furthermore, a small fraction of plant species have their genome sizes estimated (*Pellicer & Leitch, 2020*), and the cpDNA can vary substantially between species and/or of total extracted DNA (*Dodsworth et al., 2019*). Therefore, information on how shallow the low copy part of the nuclear genome is often lacking in genome skimming studies. A practical implication is that after skimming the plastome and few other regions traditionally used, the remaining and overwhelming amount of DNA sequenced is usually not analyzed and wasted. Most studies that aimed to recover low copy nuclear markers from genome skims have relied on reference-guided approach, with varied strategies for data filtering (*Besnard et al., 2018*; *Olofsson et al., 2019*; *Vargas et al., 2019*; *Loiseau et al., 2021*; *Meng et al., 2021*). Nonetheless, a user-friendly pipeline to perform the whole process, from data assembly to alignment generation, is still lacking.

In this article, a pipeline and its companion R package designed to automate the recovery of low copy nuclear markers from genome skimming libraries are presented. The pipeline includes steps to map reads to references, then generate consensus sequences, and single loci and concatenated alignments for phylogenetic use. The R package includes functions for alignment filtering and basic sequence statistics. Using an empirical data set I

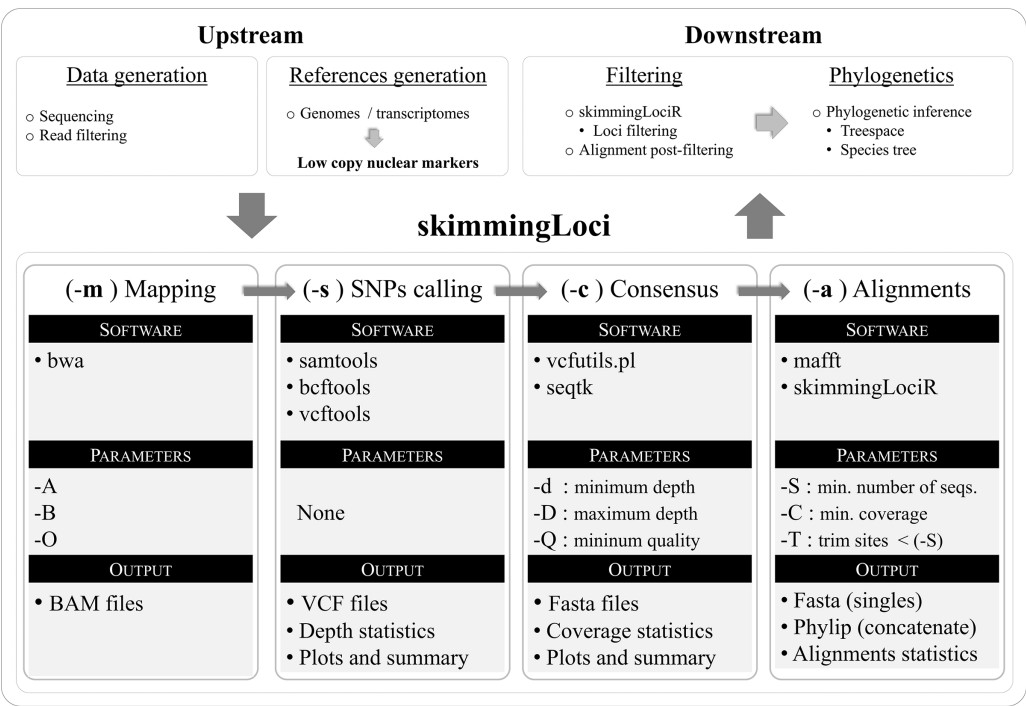

**Figure 1** Flowchart illustrating key steps and software used in the skimmingLoci pipeline, as well as in downstream and upstream major steps.

explore the effect of reference selection and key parameter settings in this pipeline. Given the low depth, low coverage and high amount of missing data that will likely be associated with attempts to harvest low single copy markers from genome skimming data a series of analyses were designed to evaluate the impact of such characteristics in the phylogenetic outcomes. To further validate this approach two published phylogenetic data sets (one low copy gene—Sanger sequencing; and one target enrichment—high throughput sequencing) were assembled through this pipeline and analyzed.

## MATERIALS AND METHODS

### Pipeline overview

The pipeline was written in bash and a flow chart illustrating its key steps is available in Fig. 1. In order to use the pipeline, the user is required to provide filtered reads and a reference file in fasta format (including markers to be assembled). The pipeline performs an automated reference-based assembly process for one or several libraries, including four key steps: mapping (–m), SNPs calling (–s), consensus generation (–c), and alignment generation (–a). The companion R package skimmingLociR includes functions to perform post-filtering steps and generate basic alignment descriptors. A tutorial is available in https://github.com/mreginato/skimmingLoci.

In this pipeline "depth" refers to the number of aligned reads supporting a base call, while "coverage" refers to the relative amount of base pairs recovered in the consensus sequence in relation to the reference (completeness). The mapping step uses the software

bwa (*Li & Durbin, 2009*) and options from this program available in the pipeline to be modified relates to matching score, and mismatch and gap penalties (–A, –B, –O). This step generates files in bam format that are used in the next step. The SNP calling step is performed with vcftools (*Danecek et al., 2011*), and functions from samtools (*Li et al., 2009*) and bcftools (*Li, 2011*) are used in intermediary steps. This step generates files in VCF format and depth statistics (plots and summaries). Sequence consensus generation based on the VCF files is performed with seqtk (https://github.com/lh3/seqtk), and vcfutils.pl (*Li et al., 2009*) is used in intermediate steps. Consensus sequences are generated in fasta format, and coverage statistics are also provided (plots and summaries). The last step in the pipeline is the alignment generation, intermediate steps are performed with internal function of skimmingLociR package and alignment with mafft (*Katoh & Standley, 2013*).The companion R package skimmingLociR includes functions for alignment post-filtering and handling, such as wrappers to generate basic alignment descriptors (alignStats), extract SNPs from an alignment (extractSNPs), concatenate lists of alignments (fastConc), trim and fill alignment edges (fillAlignments and trimAlignments), and filter loci (filterLoci). Most of these functions call internally several functions from the R package ape (*Paradis & Schliep, 2019*). The pipeline, the companion R package, and help files are available at https://github.com/mreginato/skimmingLoci. Control files with commands and parameters used in the assemblies, intermediate data files, and script used in this study are also available (https://doi.org/10.5281/zenodo.7157236).

## Genome skimming data

Genome skimming libraries used across all assemblies are the same used to generate full plastome sequences of Melastomataceae (*Reginato et al., 2016*). Sampling included 16 species across major clades in the family. Genome size is unknown for these species and for the genera they belong. Voucher information and details about DNA extraction and sequencing are available in the original publication. Libraries included paired-reads with length of 100 base pairs. Prior to all assemblies reads were quality trimmed at 0.05 probability and filtered by length (<50 bp removed) in Geneious 7.1 (Biomatters Ltd., Auckland, New Zealand). All assemblies presented in this article included the same 16 quality filtered libraries.

## References comparison

To evaluate the effect of reference selection (conserved *vs.* less conserved sequences) in the pipeline output two different reference sets were assembled with the same assembling parameters (minimum depth: –d = 2; minimum coverage: –C = 0.1; with alignment edge trimming enabled: –T). References included filtered transcripts (CDS), and full genes including exons and introns (full). Two Melastomataceae transcriptomes (*Tetrazygia biflora* (Cogn.) Urb. and *Medinilla magnifica* Lindl.) were downloaded from the onekp.org database (*Leebens-Mack, Wong & One Thousand Plant Transcriptomes Initiative, 2019*) and used to build the reference sets. Putative low copy nuclear markers within the sequenced transcriptomes were identified with the MarkerMiner pipeline (*Chamala et al., 2015*). Parameters were left as default and the minimum transcript length was set to 400.

The pipeline identified 949 CDS of putative low copy genes which were kept and further processed. In order to recover the full sequence of the 949 candidate genes (*i.e.*, including introns) the Melastomataceae CDS were imported into Geneious 7.1 (Biomatters Ltd., Auckland, New Zealand) and a series of mapping and *de novo* assemblies were conducted. This process included a mapping step of all reads to the references (949 CDS), save the mapped reads, perform a *de novo* assembly using the saved reads, and mapping the resulting contigs back to the original references. This process was repeated several times until no progress was detected. Both mapping and *de novo* assembly were performed with Geneious algorithms. Mapping was performed with the high sensitivity settings with the "maximum gap size" option set to 1,000, *de novo* was performed with default options. A total of 683 genes were fully recovered in this process and constitute the full reference set (full; 1,905,815 bp). The same 683 genes were identified among the output of MarkerMiner and were selected to build the CDS reference set (CDS; 985,008 bp).

Additionally, for each of the two reference sets assemblies two different post-filtering programs were used to remove putative poorly aligned sites within the individual loci alignments. Moderate filtering was performed with Gblocks v.0.91b (*Castresana, 2000*) using the following parameters: b1 = 70%; b2 =70%; b3 =100%; b4 =10; b5 = "all". A second stronger filtering scheme was achieved with Aliscore.pl v.2.0 (*Misof & Misof, 2009*), where the options "–N –r –i" were enabled. Thus, six assemblies were compared in this step: CDS (with no-filtering, with moderate filtering, and strong filtering), and full genes (with no-filtering, with moderate filtering, and strong filtering).

Unless otherwise stated, all analyses and plots were generated in R 3.4.0 (*R Development Core Team, 2016*). Assemblies' comparison included the following metrics for individual loci in each assembly: length of the sequence; coverage (median); depth (median), number of variable sites, number of parsimony informative sites (PIS), and missing data percent. Metrics tabulated for the concatenated alignment of all loci included the median bootstrap support value in its phylogenetic tree and the RF distance (*Robinson & Foulds, 1981*) of the concatenated tree to the full plastome tree published in *Reginato et al. (2016)*. Tree distance was calculated with the R package phangorn (*Schliep, 2011*). Concatenate tree inference was performed with Maximum Likelihood in RAxML v.8.2.4 (*Stamatakis, 2014*). The GTR+G model was employed and support was estimated through 100 bootstrap replicates.

Additionally, to evaluate the impact of read number across libraries in the pipeline output, correlations were performed between total number of reads and: total base pairs assembled, median depth across individual loci, and median coverage across individual loci. Pearson's product-moment correlation was implemented with the cor.test function of the R package stats (*R Development Core Team, 2016*). For these analyses the "full" reference set was used (parameters –d 2 –C 0.1) with no post-filtering.

## Key parameters comparison

To evaluate the effect of key parameters selection (minimum depth and minimum coverage), a comparison of assemblies with the same reference set (full genes) was performed. A total of eight assemblies were generate. In four assemblies, minimum depth

(–d) was kept at two and the minimum coverage (–C) was set to 0.1, 0.3, 0.5, and 0.7; while in the other four assemblies the minimum coverage (–C) was kept at 0.1 and the minimum depth (–d) was set to 2, 3, 4, and 5. Assemblies' comparison included the same metrics tabulated for the previous comparison of different reference sets. No alignment post-filtering was applied in these assemblies.

### Depth, coverage, missing data and outliers

Giving the low depth, coverage and high amount of missing data that will likely be associated with studies using this pipeline, a series of analyses were designed to evaluate their impact in the phylogenetic outcomes. For the following analyses the assembly using the full reference set with the following parameters was used: –d 2 –C 0.1 –T. No alignment post-filtering was applied in this assembly. In order to identify the impact of alignment completeness (missing data percent and median coverage), as well as other characteristics of individual loci that might influence phylogenetic inference (such as number of base pairs), correlations between these metrics and mean gene tree bootstrap support across all individual loci were determined. First, pairwise Pearson's product-moment correlation between all predictors were conducted as previously described, including number of variable sites, total number of aligned base pairs, PIS, missing data percentage, median coverage, median coverage standard deviation and median depth. Representative uncorrelated variables were selected for the next analysis, where redundant variables with Pearson's r > 0.7 were not considered. The effect of the uncorrelated predictors (total number of aligned base pairs, missing data percentage, coverage standard deviation, and median depth) on mean bootstrap support across all gene trees was then assessed through multiple linear regression implemented in R. Metrics other than ratios were log transformed prior to all analyses.

Additionally, the total number of base pairs, median depth, mean coverage, missing data percentage, mean bootstrap, and quartet distance to the concatenate tree between putative outlier loci and the remaining loci were compared and significance assessed with Wilcoxon tests implemented in R. *P*-values were adjusted with the p.adjust function using Holm's method (*Holm, 1979*). Outlier loci were identified with a treespace analysis. Gene trees for each locus were estimated in RAxML (as previously described) and a distance matrix (quartet distance) including all gene trees pairs was constructed using the R package Quartet 1.1 (*Smith, 2019*). Since different loci might include different samples, when necessary unmatched terminals were dropped in each pair under calculation. The distance matrix was then subjected to a Principal Coordinates Analysis with the R package ade4 (*Dray & Dufour, 2007*). Outlier loci were identified with the Mahalanobis distance (*p*-value < 0.05) based on the first three axes implemented with mahalanobisQC from the R package ClassDiscovery (*Coombes, 2019*).

### Species tree

In order to evaluate whether alternative phylogenies might also relate to the inference method a species tree analyses was performed for comparison with the concatenate ML tree. Species tree was inferred using Astral v 5.6.3 (*Zhang et al., 2018*), with default options

and support was estimated with gene bootstrapping (–gene-only option). Species tree inference was based on the 683 gene trees estimated with RAxML (as previously described) from the assembly using the full reference set (–d 2 –C 0.1 –T, no alignment post-filtering).

## Assembly of published data sets

To further validate this pipeline a single low copy loci, the nuclear gene that encodes the chloroplast-expressed glutamine synthetase (*ncpGS*), for which a Sanger-based phylogeny is available (*Ionta et al., 2007*) and a target enrichment data set (Angiosperm353 probe set) for the Myrtales (*Maurin et al., 2021*) were assembled. Both data sets were included because they share species in common with the skimming libraries analyzed here (*Rhexia virginica* L. for the *ncpGS*, and nine species for the target enrichment). The target enrichment data set (Myrtales) was downloaded from http://sftp.kew.org/pub/paftol/, where only Melastomataceae plus its sister clade (CAP) were kept. The longest sequence per individual alignment was selected and used as reference in the assembly pipeline (totaling 344 loci). Both data sets were assembled with a minimum depth of 2 and a minimum coverage of 0.1. The resulting assembled sequences were then re-aligned with the original published data sets. Sequences were aligned with MAFFT v.7 using the FFT-NS-i strategy (*Katoh & Standley, 2013*). The ML trees were estimated with RAxML as previously described. For the target enrichment assembly, the number of loci and coverage for each sample assembled with skimmingLoci was compared with the observed in the published data set. Comparisons (skimmingLoci assembly *vs.* published samples) were performed with Wilcoxon rank sum test in R.

## RESULTS

The 16 libraries analyzed throughout this article have a total number of reads ranging from ca. 4 to 22M reads. The number of recovered loci, total base pairs, median depth and median coverage for the full assembly (–d 2 –C 0.1, no alignment post-filtering) are presented in Table 1. Only in one library all loci were at least partially recovered (*M. dodecandra*). On average, 618 loci (ca. 90% of target loci) with around 1,000,000 base pairs (ca. 57% of target base pairs) were recovered across the libraries. Individual loci median depth was usually low across libraries (median = 2, s.d.= 1.36), while the median coverage was 0.39 (s.d.= 0.21), indicating that in most cases loci were only partially recovered. A moderate correlation was observed between the total number of reads and median depth (r = 0.65, *p*-value = 0.007); as well as between the total number of reads and mean coverage (r = 0.50, *p*-value = 0.047), indicating that libraries with higher number of reads tend to yield more assembled data (plots available in Fig. S1). Nonetheless, the correlation between number of reads and total base pairs recovered was lower (r = 0.30, *p*-value = 0.254). While there is an overall trend, some samples had a relatively greater yield (*B. schlimii*), while a few showed a relatively lower output (*R. bracteata*; Table 1; Fig. S1).

**Table 1 Summary characteristics of the 16 libraries assembled using the "full" reference set (parameters –d 2 –C 0.1).**

| Species | Reads | Loci (n) | Total (bp) | Coverage (median, s.d.) | Depth (median, s.d.) |
|---|---|---|---|---|---|
| *Allomaieta villosa* (Gleason) Lozano | 22,583,550 | 682 | 1,486,675 | 0.72 (0.14) | 5 (10.28) |
| *Bertolonia acuminata* Gardner | 18,820,316 | 682 | 1,433,836 | 0.61 (0.13) | 3 (6.04) |
| *Blakea schlimii* (Naudin) Triana | 6,272,448 | 682 | 1,473,735 | 0.6 (0.16) | 3 (5.37) |
| *Eriocnema fulva* Naudin | 2,369,052 | 614 | 953,093 | 0.2 (0.09) | 1 (1.92) |
| *Graffenrieda moritziana* Triana | 18,255,622 | 683 | 1,646,960 | 0.74 (0.13) | 5 (16.22) |
| *Henriettea barkeri* (Urb. & Ekman) Alain | 3,904,930 | 621 | 993,742 | 0.25 (0.11) | 1 (4.76) |
| *Merianthera pulchra* Kuhlm. | 7,262,788 | 571 | 792,362 | 0.23 (0.14) | 1 (3.11) |
| *Miconia dodecandra* Cogn. | 14,915,062 | 683 | 1,820,433 | 0.76 (0.11) | 4 (11.04) |
| *Nepsera aquatica* (Aubl.) Naudin | 14,750,648 | 682 | 1,034,413 | 0.47 (0.17) | 3 (16.74) |
| *Opisthocentra clidemioides* Hook. f. | 6,985,796 | 676 | 1,159,469 | 0.39 (0.13) | 2 (3.52) |
| *Pterogastra divaricata* (Bonpl.) Naudin | 8,998,186 | 659 | 874,845 | 0.34 (0.15) | 2 (6.66) |
| *Rhexia virginica* L. | 12,157,014 | 674 | 967,659 | 0.38 (0.15) | 2 (17.31) |
| *Rhynchanthera bracteata* Triana | 22,213,604 | 528 | 596,533 | 0.21 (0.11) | 2 (10.62) |
| *Salpinga maranoniensis* Wurdack | 14,197,808 | 478 | 699,934 | 0.19 (0.11) | 1 (15.92) |
| *Tibouchina longifolia* (Vahl) Baill. | 9,425,454 | 682 | 994,943 | 0.42 (0.16) | 3 (7.2) |
| *Triolena amazonica* (Pilg.) Wurdack | 6,664,094 | 286 | 508,254 | 0.14 (0.07) | 1 (3.86) |

**Note:**
Reads, total number of reads in each library; Loci (n), number of recovered loci; Total (bp), total based pairs recovered; Coverage (median, s.d.), median and standard deviation coverage across individual loci; Depth (median, s.d.), median and standard deviation depth across individual loci.

## Assemblies with distinct references

Reference selection and alignment post-filtering strategy impact was evaluated through comparisons of two different reference sets (CDS and full), each one with three aligned base pairs post-filtering schemes (no-filtering, moderate filtering, and strong filtering). Summary statistics of each of these six assemblies are presented in Table 2. The relative total number of aligned base pairs recovered was higher in the CDS (90%) than in the full reference set (80%), indicating that the reference including only more conserved base pairs (exons) had a relatively higher yield. On the other hand, the full reference set (with the highest number of target base pairs) also resulted in the higher number of total base pairs, percent of missing data and mean bootstrap support in the concatenated tree, suggesting that more data is preferable for a higher bootstrap support, despite potential increase in missing data.

Alignment post-filtering decreased mean bootstrap support in the concatenate tree of the CDS reference set (Table 2). The full reference set comparisons had similar mean bootstrap values, with moderate alignment post-filtering slightly increasing mean bootstrap support, while strong filtering resulted in a small decrease (Table 2). While the mean bootstrap support did not change strongly across filtering schemes, the resulting length of the alignment was highly affected. For instance, strong filtering in the full reference left only 25% of the original data set. The same trend is observed for the other reference set, indicating that amount of data excluded by post-filtering strategies was large, but mean bootstrap support was not highly affected.

**Table 2 Summary statistics of each of the six assemblies analyzed, with distinct references ("full" and "CDS") and three different levels of alignment post-filtering ("none", "moderate", and "strong").**

| Reference/filtering | Loci (n) | Aligned (bp) | Variable | PIS | Missing data (%) | Bootstrap (mean) |
|---|---|---|---|---|---|---|
| Full/none | 683 | 1,532,601 | 472,022 | 121,647 | 49.6 | 95.5 |
| Full/moderate | 682 | 1,313,414 | 413,152 | 108,389 | 47.8 | 96.5 |
| Full/strong | 683 | 377,028 | 117,781 | 38,952 | 30 | 95 |
| CDS/none | 683 | 885,828 | 242,192 | 75,228 | 44.9 | 95.9 |
| CDS/moderate | 676 | 690,926 | 194,921 | 63,410 | 40.9 | 93.8 |
| CDS/strong | 683 | 298,568 | 85,957 | 29,075 | 31.4 | 92.9 |

**Note:**
Loci (n), number of loci; Aligned (bp), total number of aligned base pairs; Variable, total number of variable sites; PIS, total number of parsimony informative sites; Missing data (%), Total percent of missing data; Bootstrap (mean), mean boostrap in the concatenated phylogeny.

Despite the variation in total number of reads, missing data, and bootstrap support across the three different schemes within the same reference set, the recovered topologies were relatively stable across comparisons. CDS assemblies resulted in the same topology across all different filtering schemes. A topology similar to the recovered in the CDS assemblies was also recovered for the strong filtering scheme of the full reference assembly, with one distinction involving the relationship of *M. pulchra*. The other two filtering schemes of the full reference set had yet an additional distinct relationship involving *B. schlimii*. The recovered phylogenies with support information of the six assemblies are available in Fig. S2.

## Assemblies with distinct key parameters

Key parameters selection, including the minimum depth to keep a base call in the consensus sequence (–d) and the minimum coverage of a sequence to be included in the final locus alignment (–C), were evaluated through comparisons of eight assemblies where these parameters were set to vary. Making both parameters more conservative resulted in a similar pattern (Figs. 2A, 2C, Table 3). The number of recovered loci, total aligned base pairs, variable sites, and PIS have consistently decreased with more stringent settings, while the relative percent of missing data was increased (Table 3). In the more conservative minimum coverage setting (–d 2 –C 0.7), on average, only 19% of the target loci and 28% of target base pairs were partially recovered, and all sequences of three libraries were totally removed (*i.e.*, resulting in 13 out of 16 samples in the concatenate alignment). Mean bootstrap support in the concatenate tree deviated from the general pattern observed in the other metrics. In both parameters comparisons, the bootstrap support had an initial increase followed by an abrupt decrease (Figs. 2B, 2D). These results indicate that small changes in these parameters have a high impact in the assembly outcome. Additionally, parameter tuning through assembly tests with different settings might increase desired features in the phylogenetic outcome (*e.g.*, higher bootstrap support).

Concatenate tree topologies across the eight assemblies were reasonably similar, despite greater variation in bootstrap support (Figs. S3 and S4). These analyses included only the

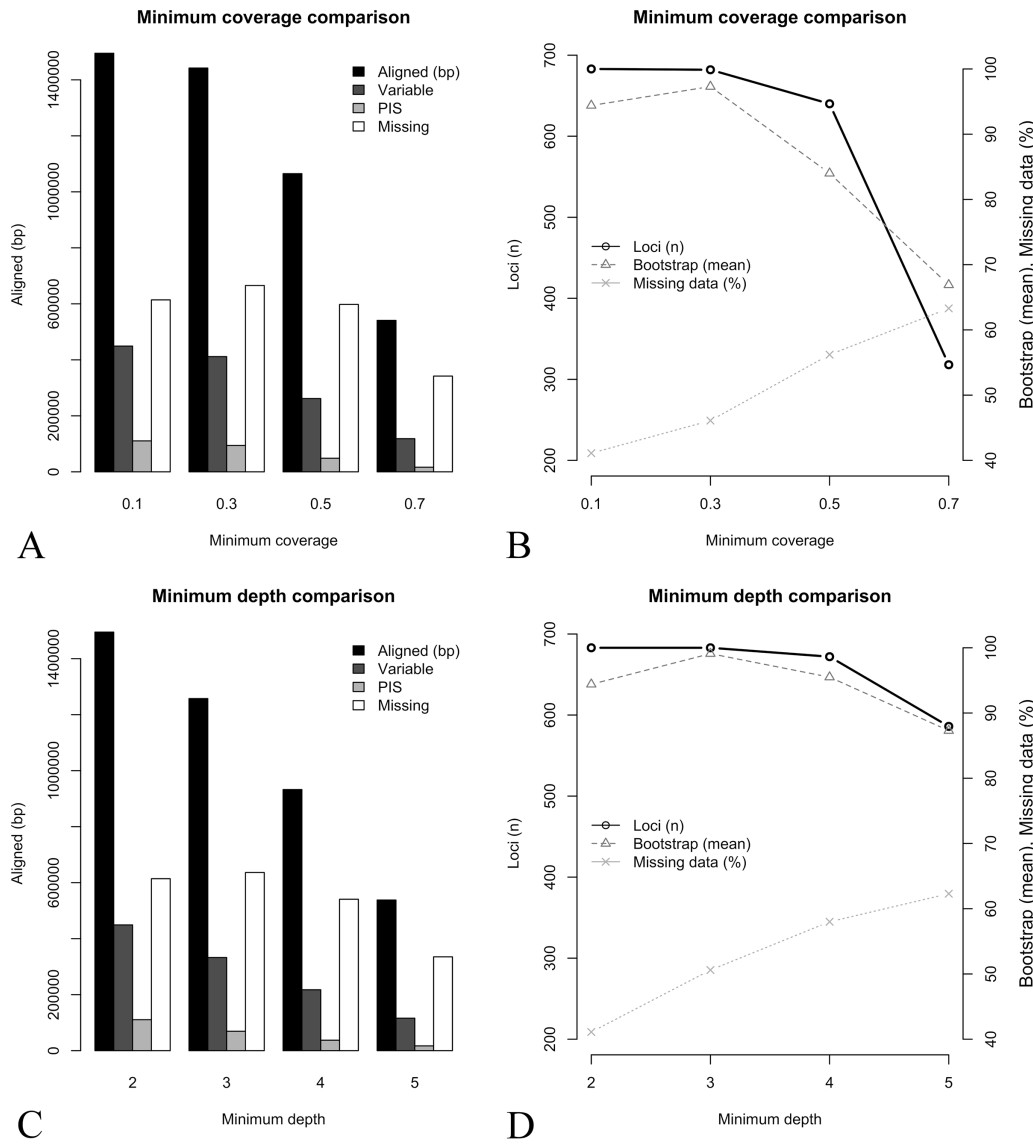

**Figure 2 Key parameters comparisons, including the minimum depth to keep a base call in the consensus sequence (–d parameter) and the minimum coverage of a sequence to be included in the final locus alignment (–C parameter).** (A) Minimum coverage *vs.* aligned base pairs (bp), variable sites (Variable), parsimony informative sites (PIS), and missing data (Missing). (B) Minimum coverage (–C) *vs.* number of loci (Loci n), mean bootstrap support (Bootstrap mean) and percent of missing data (Missing data %). (C) Minimum depth (–d) *vs.* aligned base pairs (bp), variable sites (Variable), parsimony informative sites (PIS), and missing data (Missing). (D) Minimum depth (–d) *vs.* number of loci (Loci n), mean bootstrap support (Bootstrap mean) and percent of missing data (Missing data %).

13 libraries in common across all comparisons, where changes in both parameters resulted in the same pattern of phylogenetic conflict. For both minimum depth and minimum coverage, the concatenate trees presented the same discordant relationships previously observed in the reference sets comparisons (involving the position of *B. schlimii*, the position of *M. pulchra* and the relationship of *R. bracteata* and *N. aquatica*).

**Table 3 Summary statistics of each of the eight assemblies analyzed, with distinct values of key parameters including the minimum depth to keep a base call in the consensus sequence (–d) and the minimum coverage of a sequence to be included in the final alignment.**

| Key parameters | Loci (m, r) | Terminals | Aligned (bp) | Variable sites | PIS | Missing data (%) | Bootstrap (mean) |
|---|---|---|---|---|---|---|---|
| –d 2 –C 0.7 | 120 [1, 313] | 13 | 540,764 | 118,386 | 16,592 | 63.3 | 66.9 |
| –d 2 –C 0.5 | 268 [8, 638] | 15 | 1,064,863 | 261,671 | 48,638 | 56.2 | 84 |
| –d 2 –C 0.3 | 423 [18, 682] | 16 | 1,442,185 | 411,563 | 93,974 | 46.1 | 97.3 |
| –d 2 –C 0.1 | 618 [286, 683] | 16 | 1,494,809 | 449,182 | 110,495 | 41.1 | 94.4 |
| –d 3 –C 0.1 | 490 [39, 683] | 16 | 1,257,452 | 332,727 | 69,297 | 50.6 | 99.1 |
| –d 4 –C 0.1 | 382 [8, 672] | 16 | 932,497 | 217,549 | 37,358 | 58 | 95.5 |
| –d 5 –C 0.1 | 287 [1, 586] | 16 | 537,866 | 115,958 | 17,078 | 62.3 | 87.3 |

Note:
Loci (m, r), mean and range number of recovered loci (at least partially) across all terminals; Terminals, Number of terminals in the concatenated alignment; Bootstrap (mean), mean boostrap in the concatenated phylogeny (including the same 13 terminals).

## Depth, coverage, missing data and outliers

A group of seven descriptors, with emphasis on alignment completeness and informativeness, were selected to evaluate their impact on mean bootstrap support across individual loci gene trees. Pairwise correlations revealed that some descriptors were strongly correlated and formed five groups (Fig. S5). The first group is related to informativeness and included the total number of base pairs, number of variable sites and number of PIS, while the second group related to alignment completeness included missing data percent and mean coverage (Fig. S5). Median depth and the coverage standard deviation were not strongly correlated with any other descriptor (Fig. S5). Then, the effect of the selected uncorrelated descriptors on mean bootstrap support was evaluated with multiple linear regressions. The resulting model had an adjusted $R^2$ of 0.37 ($p$-value $< 2.2e{-}16$) and the relative importance of predictors were: total number of base pairs = 93.3%; Median depth = 4.6%; Coverage standard deviation = 1.8%; and missing data percent = 0.3%. This result indicates that most of the variation explained by the predictors is related to the length of the alignment (informativeness), and missing data (alignment completeness) has very low predictive power on bootstrap support across gene trees (plots of individual predictors on Figs. 3H–3K).

In order to further compare alignment completeness and informativeness impact on gene trees, a treespace analysis was performed and putative outlier loci identified. A total of 63 out of 683 (ca. 9%) loci were flagged as outliers (Fig. 3A). The distribution of the descriptors in the outlier loci and in the remaining ones is shown in Figs. 3B–3G. The total number of base pairs, median depth, and mean bootstrap support were significantly lower in the outlier loci when compared to the remaining ones, while the opposite was verified for the quartet distance to the concatenate tree ($p$-value $< 0.05$). Missing data percent and mean coverage did not show significant difference between the two groups (Figs. 3B–3G).

## Species tree

The species tree inferred with Astral presented high gene bootstrap support along most nodes (Fig. 4). Exceptions include moderate support for the placement of *N. aquatica*, and low support for *M. pulchra*. The overall topology is similar to the one observed in the

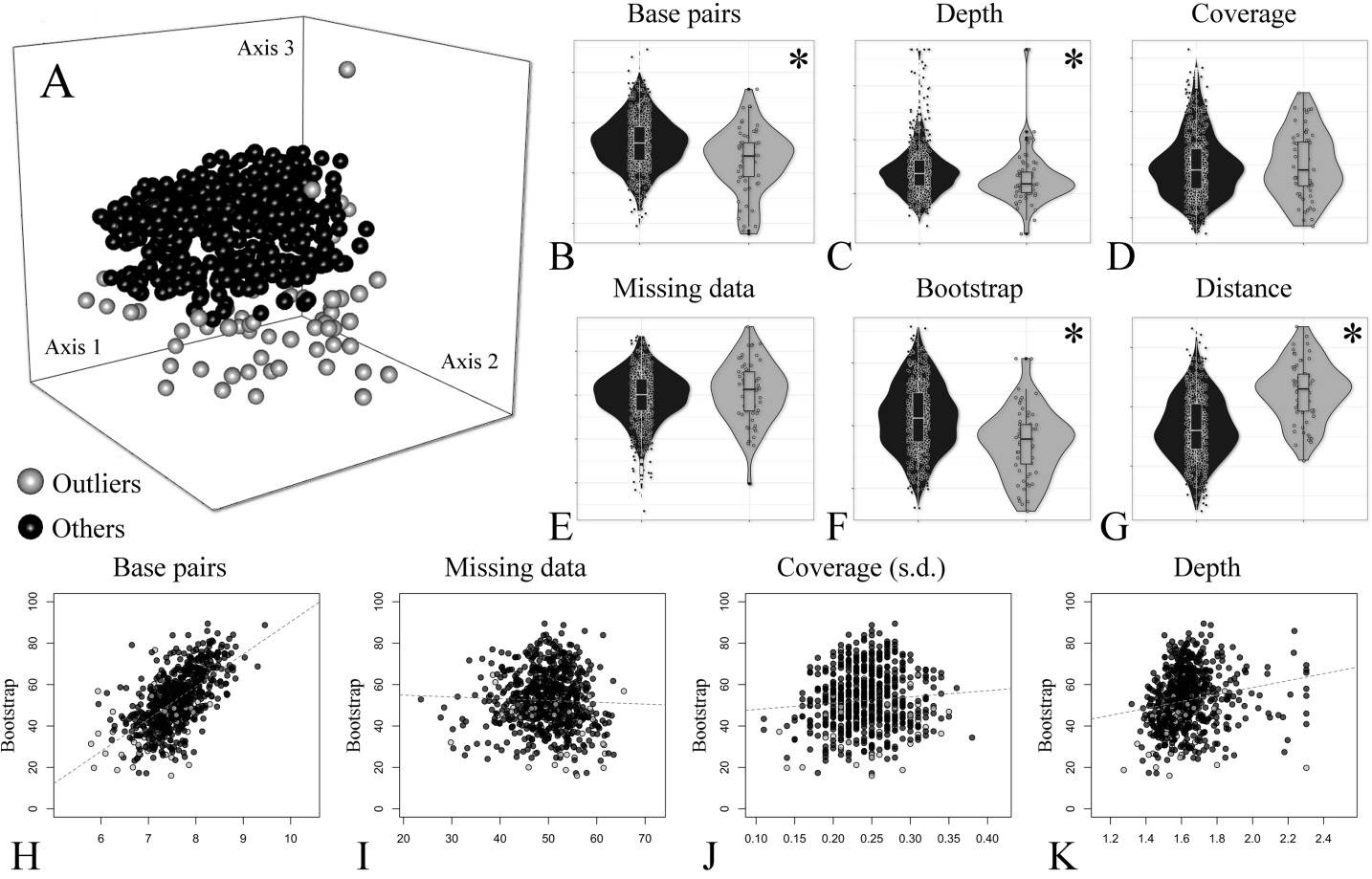

**Figure 3 Treespace and comparative descriptors of outlier loci and the remaining ones.** (A) Treespace analysis indicating putative outlier loci identified (63 out of 683 loci were flagged as outliers). (B–G) Descriptors distribution comparison between outlier loci and in the remaining ones (violin plots). (B) Total base pairs. (C) Median depth. (D) Mean coverage. (E) Missing data. (F) Mean bootstrap. (G) Distance (RF) to the concatenate tree. (H–K) Biplots of selected descriptors *vs*. mean bootstrap support. (H) Total base pairs. (I) Missing data percent. (J) Coverage standard deviation. (K) Median depth. In all plots outliers are shown in gray and the remaining loci in black. The asterisk (*) indicates significant difference between groups.

concatenate ML tree (Fig. 4), with the exception of the uncertain relationships above mentioned and the relationship of *B. schlimii*. These conflicts are the same incongruences observed in the reference set and key parameters comparisons, indicating that to some extent the conflict observed across the comparisons with different reference sets and key parameters might be related to gene tree conflict.

## Assembly of published data sets

Assembly of the *ncpGS* locus of *Rhexia virginica* resulted a sequence with median depth of 3.3 and a coverage of 0.49. Alignment with the original data set included a total of 462 base pairs, 65 variable and 10 parsimony informative sites, and 8% of missing data (Fig. S6). Overall, the maximum likelihood tree showed low support across most nodes, but the genome skim terminal of *R. virginica* grouped with the Sanger-based terminal of the same species (Fig. S6).
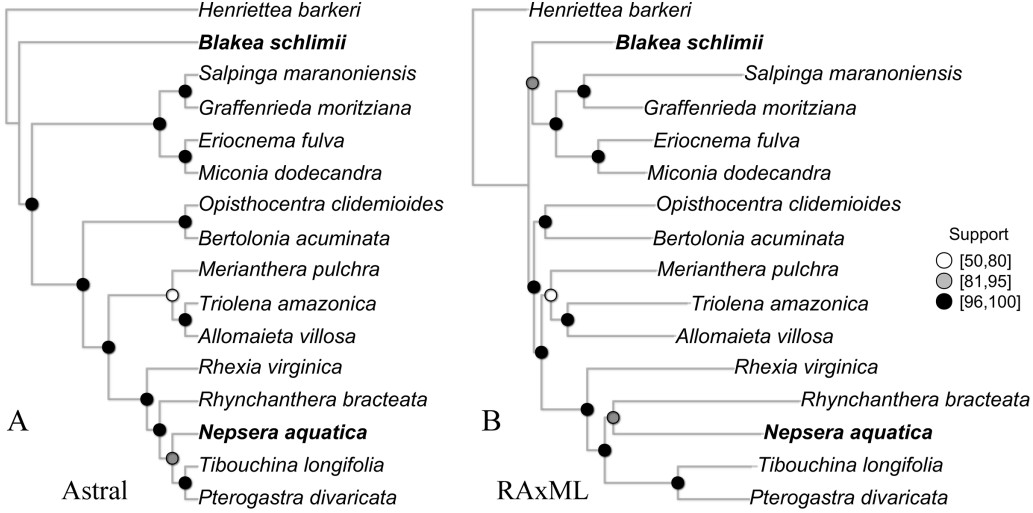

**Figure 4 The species tree inferred with Astral (A) and the maximum likelihood tree of the concatenate alignment (B).** Both trees from the "Full" assembly (−d 2, −C 0.1). Support values are depicted following the legend (A, Gene bootstrap; B, Bootstrap). Terminals with distinct phylogenetic positioning in bold face.

Assembly of the target enrichment data set (Myrtales, Angiospers353 prose set) included nine genome skimming samples and had a median depth of 2. Out of the 344 loci, genome skimming libraries recovered a mean of 318 loci (ranging from 144 to 336) *vs.* 323 (240 to 344) in the target enriched libraries. Median coverage across genome skimming libraries was 0.53 (ranging from 0.16 to 0.84), while target enriched samples had a median coverage of 0.52 (0.19 to 0.87). No significant difference was observed when compared both the number of loci and median coverage between genome skimming libraries with target enriched ones (number of loci *p*-value = 0.23; median coverage *p*-value = 0.60). The maximum likelihood tree including the published terminals along with the skimmingLoci assemblies is available in Fig. 5. Most genome skimming libraries were recovered as sister to the same species of target enrichment samples (Fig. 5), the only exception was the *Eriocnema fulva* library that was recovered near, but not sister to the other sample of this species, but with no support. *Eriocnema fulva* was among the three libraries with the lowest number of loci and median coverage (the other two were *Triolena amazonica* and *Salpinga maranoniensis*; Fig. 5).

# DISCUSSION

Despite a few attempts to use genome skims to recover low copy nuclear loci for direct phylogenetic use (*Besnard et al., 2014*; *Besnard et al., 2018*; *Olofsson et al., 2019*; *Vargas et al., 2019*; *Liu et al., 2021*; *Loiseau et al., 2021*; *Meng et al., 2021*; *Cai, Zhang & Davis, 2022*), such endeavor remains largely neglected. Causes might include shallow depth of the low copy part of the genome due to libraries with few reads, species with large genomes, and especially, the trade-off between these two, but also might be related to the lack of

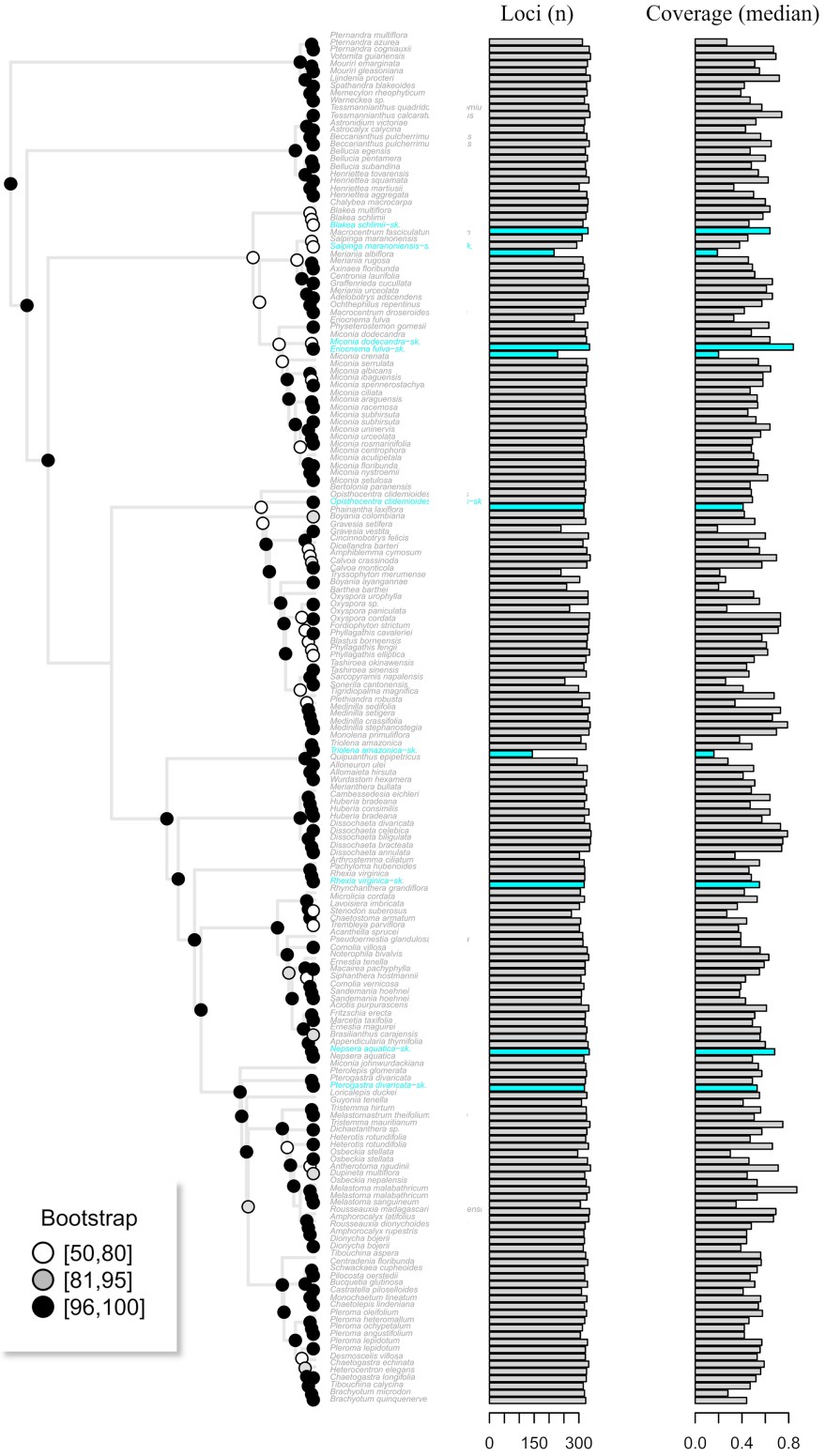

Figure 5 The maximum likelihood tree of the target enrichment data set (Myrtales, Angiospersm343 probe set) including the published terminals along with the skimmingLoci assemblies (in blue). The total number of loci and median coverage for each terminal are plotted on the right side. Bootstrap support is depicted at the nodes following the legend.

pipelines for data assembling. Pipelines commonly used with genome skimming data focus on the recovery of the plastome sequence (*Dierckxsens, Mardulyn & Smits, 2017*; *McKain & Wilson, 2017*), the major source of phylogenetic information generated in genome skimming studies. Nonetheless, the nuclear genome harbors significant information relating to variation within and among plant species, and it is decisive for a more effective identification of multiple genome donors in lineages with a historical of hybridization and allopolyploid (*Zimmer & Wen, 2013*). Furthermore, the necessity of multiple gene trees for more accurate species tree inference is now widely acknowledged. As a result, approaches such as HYB-seq, that aims at the repetitive component of the genome as well as target a portion of the low copy part, are becoming increasingly popular (*Dodsworth et al., 2019*).

Early simulations of plant total genomic sequencing have demonstrated that even at the lowest values of sequencing depth, reads originating from single-copy nuclear loci were still detected (*Straub et al., 2012*). This expectation was confirmed here with an empirical data set, where a substantial amount of putative low copy nuclear DNA was assembled and proved useful to base phylogenetic inference across the libraries tested. Depending on the settings and reference set used, total loci number partially recovered varied from 47% to 100% of the total 683 target loci, ranging from 537,866 to 1,532,601 aligned base pairs (Tables 2 and 3). Plastome data previously assembled with the same libraries rendered an alignment of 140,649 base pairs (*Reginato et al., 2016*), ca. 4 to 11 times less data depending on the assembly generated. The approach presented here was further validated for a single locus (*ncpGS*) for which a Sanger-based phylogeny was available (*Ionta et al., 2007*), where the library assembled through this pipeline clustered with the Sanger-based terminal of the same species. The same was observed with the assembly of the target enrichment data set (Myrtales, Angiosperms353 probe set), where all but one library, clustered with the same species of the original data set. The only exception (*E. fulva*) was among the libraries that yielded the smallest amount of assembled data.

Genome skimming was conceived as a gDNA shallow sequencing method (*Straub et al., 2012*). As a result, it is expected that most loci of the low copy part of the genome will not be fully covered, neither will present high depth. When dealing with deep sequenced regions in genome skims (such as the plastome), it is common practice to use *de novo* methodologies (*Dierckxsens, Mardulyn & Smits, 2017*; *McKain & Wilson, 2017*). However, similar strategies will likely be inefficient for the low copy component, giving its fragmented nature and lower depth. Thus, mapping methods are an efficient alternative, but they require a reference to anchor the reads during the procedure. In addition to reference selection, other critical aspects of assembling low copy nuclear markers from genome skims include key parameters (minimum coverage and depth of a sequence to be used), as well as alignment completeness. The latter is a result of the lack of coverage and/or depth across loci in different libraries, and its level might be directly linked to parameter values applied during assembly. More stringent values of minimum depth and coverage reduces the amount of assembled data and increases the relative amount of missing data (Table 3), which in turn can compromise phylogenetic inference. These issues are discussed in the sections that follow.

## Reference selection

Transcriptomes are now the major source of information to reference building for either probe development or harvesting loci in genome skims for most angiosperm lineages that still lack a close relative with a fully sequenced genome (*Chamala et al., 2015*). A limitation of this approach is that it only includes coding regions, sometimes with limited phylogenetic information at shallow inferences. This issue is alleviated by the fact that intronic and intergenic regions flanking target exons (splash zone) are usually also recovered (*Johnson et al., 2016*). Here, the amount of data generated and phylogenetic support were compared between one reference set including only coding regions (CDS) and the same loci also including introns (full). As expected, the full reference set (with the highest number of target base pairs) resulted in the higher number of total base pairs and mean bootstrap support in the concatenated tree, suggesting that more data is preferable for a higher bootstrap support even if the amount of missing data is increased (Table 2). These results highlight that attempting to use genome skimming data along with transcriptome data to build references including intronic regions is highly recommended.

The relative total number of aligned base pairs recovered was higher in the CDS only reference set (90%) than in the full reference set (80%), indicating that the references including only more conserved base pairs (exons) had a relatively higher yield, but not too disparate. This is expected given that mapping success (or hybridization success in target enrichments libraries) will correlate with similarity to the references (*Johnson et al., 2019*). Nonetheless, the amount of data recovered in the full reference set was still satisfactory, despite an estimated MRCA age of ca. of 45 My (37%–55 95% HPD) for the lineages analyzed (*Reginato et al., 2020*). Here, the same parameters in the mapping step were used across all comparisons (–A, –B, –O). Fine tuning of these parameters could also be tested to further optimize assembly output.

Manually curation of individual loci alignment is no longer an option in phylogenomic studies dealing with hundreds or thousands of loci, and several tools have been developed to automatically curate alignments by removing part of them (*Ranwez & Chantret, 2020*). The debate as to whether it is better or not to filter sequence alignments prior to phylogeny inferences is still open, and a major concern is that some filtering processes may tend to remove too much of the phylogenetic signal along with phylogenetic noise (*Ranwez & Chantret, 2020*). Here, two filtering schemes (moderate and strong) were compared to a scenario with no filtering. Results indicated a great variation in total number of reads left and missing data across the three different schemes, but the recovered topologies and mean bootstrap support were relatively stable across comparisons. Judging the effectiveness of the filtering methods on real data is challenging, but patterns of discordance can help (*Mai & Mirabab, 2018*). Thus, in the particular case of the libraries compared in this study, alignment post-filtering effect was not significantly positive, since in most cases it rendered similar topologies and support, and if anything, it decreased bootstrap support in the strongest filtering scheme. On the other hand, alignment post-filtering seems to have had a positive effect for the whole plastome alignment of the same libraries (*Reginato et al., 2016*). Thus, accessing the impact of post-filtering strategies

on phylogenomic data sets is still recommended, especially, because misaligned regions impacting a single sequence may have little impact on topology, but might compromise branch length estimations (*Ranwez & Chantret, 2020*).

## Key parameters: depth and coverage

Translating the raw sequencing data into the final sequences in reference-based assemblies requires two essential steps: read mapping and genotype inference to generate a consensus sequence (*Liu et al., 2012*). At one hand, low depth sequencing always introduces considerable uncertainty into the results and makes base calling more prone to error (*Liu et al., 2012*). Thus, relaxing the minimum depth value for a base call tend to increase the amount of error. On the other hand, making such parameter more stringent will greatly reduce the amount of assembled data (Table 3 and Fig. 2), potentially hindering the use of this approach. Minimum coverage value will have a slightly different effect. Making this parameter more stringent will also greatly reduce the amount of data generated (Table 3 and Fig. 2), but changing it in the opposite direction will allow some shorter sequences within individual loci alignment, impacting gene tree inference. Here, the effect of varying both parameters were evaluated regarding the amount of data assembled and the resulting bootstrap support and showed a similar pattern. As expected, it was found that total amount of assembled data have consistently decreased with more stringent settings, while the relative percent of missing data was increased (Table 3). Nonetheless, a different pattern was found for bootstrap support. In both parameters comparisons, the bootstrap support had an initial increase followed by an abrupt decrease (Fig. 2). In this case, the higher amount of assembled data under the most relaxed settings did not resulted in higher bootstrap support, in contrast to what was found in the reference set comparison. Lower support might be associated with higher error rate under relaxed settings, as well as to an increased presence of short sequences with low information. On the other hand, making the parameters too stringent greatly reduces the amount of assembled data (Fig. 2), limiting inference power as evidenced by the lower bootstrap support. Therefore, parameter tuning through multiple comparisons tracking their effects (*e.g.*, support) is highly recommended. Despite great variation in bootstrap support across the eight assemblies compared, concatenate tree topologies were reasonably similar (Figs. S3 and S4). Discordant relationships were the same found in the reference sets comparison, indicating that the putative higher error associated with relaxed depth values had little impact in the inferred relationships.

## Alignment completeness, informativeness and outliers

Alignment completeness is a heavily debated issue in phylogenetic inference (*Wiens, 2003* and references therein). Missing data is usually assumed to be a compromising feature in phylogenetic inference, and some phylogenomic strategies are particularly prone to it (*Eaton et al., 2017*). Here, I found levels of total missing data reaching over 60% in one of the assemblies (Table 3), but comparisons with distinct references and key parameters indicate that total missing data amount was not a decisive feature impacting on bootstrap support in the concatenate analyses. To further explore the effect of missing data,

alignment completeness and informativeness were compared across individual loci alignments and their gene trees (Figs. 3H–3K). Multiple linear regression model indicated that bootstrap support across gene trees is highly affected by alignment length (number of total base pairs), with a relative importance of 93.3%, while the relative importance of missing data was negligible (0.03%). This result corroborates the other comparisons (references and key parameters), indicating that more data is preferable despite a compromise in alignment completeness. Also, it is in agreement with the expectation that longer genes will be superior for phylogenetic reconstruction (Walker et al., 2019).

Alignment completeness and informativeness was further compared between putative outlier loci (ca. 9% of assembled loci) and the remaining (Fig. 3). Outlier loci showed lower values of total number base pairs, median depth, mean bootstrap, and concordance with the concatenate tree. Missing data percent and mean coverage did not show significant difference between the two groups (Fig. 3). These results are in agreement with previous comparisons, but they also suggest that descriptors such as alignment length, mean bootstrap support and median depth should be preferred over missing data and mean coverage for individual loci filtering.

Simulations have demonstrated that reduced phylogenetic accuracy associated with incomplete alignments is caused by taxa bearing too few complete characters rather than too many missing data cells (Wiens, 2003). The libraries analyzed here presented a high variation of assembled data (Table 1), and under some stringent parameters no data was assembled for some libraries (Table 3). A moderate correlation was observed between the total number of reads and median depth (r = 0.65, p-value = 0.007), indicating that libraries with higher number of reads tend to yield more assembled data (Fig. S1). Some samples deviated from this general pattern, but the lack of information of genome sizes for the species analyzed precludes further conclusions. Regardless of the underlying causes, one important step to be considered is removing libraries with a low yield of assembled data. Such effect was not evaluated here, but has been proved to be effective elsewhere (Gates, Pilson & Smith, 2018).

## Gene tree discordance

Although largely congruent, some discordant relationships were recovered throughout the concatenate trees of the different comparisons presented here (Figs. S2, S3 and S4). Incongruence across comparison involved the same group of terminals: *B. schilimii*, *M. pulchra*, *R. bracteata* and *N. aquatica*. Interestingly, the same terminals also show discordant positioning or low support in the ML and Astral analyses of the same assembly (Fig. 4). Therefore, topologies discrepancies between comparisons including different references or key parameters values might be related to gene tree discordance. In fact, topological discordance is greater to the plastome tree (Reginato et al., 2016), than among the different scenarios presented here. Increasing taxonomic breadth is necessary to further improve phylogenomic relationships in this large clade of plants.

## CONCLUSIONS

The availability of tools and genomic data to design probes to target the low copy of genome, as well as attempts to generate universal probe sets for angiosperms (*Johnson et al., 2019*), are a recent achievement. How informative a given reference set is for a particular clade and whether to use a universal probe set or more clade-specific probes are important questions to make with budget and phylogenetic implications. One important aspect of the approach presented here is that genome skims could be used to bridge different published data sets (*e.g.*, Sanger-based, RAD-seq, target enrichment with different probe sets, *etc.*) on a super-matrix approach. Also, as previously suggested (*Vargas et al., 2019*), another putative use is to test different probe sets *in silico* with genome skims, in order to make an informed decision to maximize phylogenetic resolution in future studies. As for other nuclear low copy assembly strategies (*e.g.*, HYB-seq), undetected paralogs might also be an issue while using the approach presented here. Although this has not been directly tackled here, while using this pipeline some post-filtering strategies to flag putative paralogs presented elsewhere could also be applied (*Kates et al., 2018*; *Andermann et al., 2019*; *Zhou, Soghigian & Xiang, 2022*).

The plastid genome has so far been the most important source of data for plant phylogenetics in the era of comparative DNA sequencing (*Davis, Xi & Mathews, 2014*). Nonetheless, within the green plant species tree there is a 'cloud' of gene trees, of which the plastid genes comprise only a small fraction (*Davis, Xi & Mathews, 2014*). The pipeline presented here might stimulate the use of genome skims to recover nuclear loci for direct phylogenetic use, increasing the power of genome skimming data to resolve phylogenetic relationships, while reducing the amount of sequenced DNA that is usually ignored. The effectiveness of such approach will likely depend on the relationship of number of reads and genome size in the libraries at hand.

## ACKNOWLEDGEMENTS

I thank F. A. Michelangeli and L. Majure for suggestions in preliminary versions of the text, and Wimalanathan Kokulapalan and Liming Cai for reviewing the manuscript.

### Funding

This work was supported by the U.S. National Science Foundation (DEB-0818399 and DEB-1343612). The funders had no role in study design, data collection and analysis, decision to publish, or preparation of the manuscript.

### Grant Disclosures

The following grant information was disclosed by the authors:
U.S. National Science Foundation: DEB-0818399 and DEB-1343612.

### Competing Interests

The author declares he has no competing interests.

## Author Contributions

- Marcelo Reginato conceived and designed the experiments, performed the experiments, analyzed the data, prepared figures and/or tables, authored or reviewed drafts of the article, and approved the final draft.

## Data Availability

The data and code available at GitHub and Zenodo:

https://github.com/mreginato/skimmingLoci; Reginato, Marcelo. (2022). A pipeline for assembling low copy nuclear markers from plant genome skimming data for phylogenetic use. https://doi.org/10.5281/zenodo.7157237.

## Supplemental Information

Supplemental information for this article can be found online at http://dx.doi.org/10.7717/peerj.14525#supplemental-information.

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
