# Peer review of "A pipeline for assembling low copy nuclear markers from plant genome skimming data for phylogenetic use"

_PeerJ, doi:10.7717/peerj.14525_

## Round 0.1 · original submission · Major Revisions

Dear Author

Please consider any outstanding revision requests from all reviewers.

We encourage you to submit your revised manuscript with tracked changes to facilitate the review.

Thank you for your consideration.

·

Basic reporting

The article has been well written and understandable to a scientific audience. I do see some grammatical and sentence structure issues throughout the article, but these can be easily fixed. At times these sentences impede the reader from understanding the meaning intended by the author. The sentences have been highlighted in the accompanying PDF. Authors have provided relevant background in sufficient details and a new reader can understand the relevant background and rationale for the current study is correct. The article structure adheres to the PeerJ standards, and all the data have been shared. The article is complete and self-contained.

Experimental design

The article is within the aim and scope of Peerj journal. I am impressed with the exhaustive nature of the investigation performed by the author. The re-use of the data and developing a pipeline to extract loci from genome skimming is useful for others looking to expand this approach. The experimental design is valid, and the data-sources used for the analysis are valid in-terms of both raw data that was selected and the results that are explained in the article. The author’s idea is built from the previous work in the field and utilizes a new method to extract otherwise unused information from existing data. This approach fills an identified technical knowledge gap that makes the approach accessible.

Validity of the findings

The article is based on the pipeline hosted at https://github.com/mreginato/skimmingLoci. I cannot verify that the pipeline can be run following provided documentation.
1. Cannot install dependencies using the documentation provided. The dependencies can be installed via conda (instructions provided are missing a script) or the whole pipeline can be containerized (Docker or singularity). The authors do not have working instructions for novice users.
2. The mapping step doesn’t produce the expected even after manually installing the software tools using conda.
3. Mapping: The dependency message mentions muscle instead of mafft as mentioned in the article (line: 198)
4. Some of the parameters that can be provided are hard-coded in the commands. This is misleading the users and could yield incorrect results.
E.g. the -A, -B, -O parameters for bwa
“bwa mem $REFERENCE $WD_IN/$i.R1.fastq.gz $WD_IN/$i.R2.fastq.gz "-t$CORES" -A 1 -B 3 -O 5 -a -T 10 -M 2> $BAM_DIR/$i.bwa.log | samtools view -@$CORES -q 1 -b -F 4 -S -T $REFERENCE 2>/dev/null | samtools sort -@$CORES -o $BAM_DIR/$i.bam 2>/dev/null”

Unfortunately, I do not see a convincing argument for reviewing the results until the issues with the pipeline are fixed. This could be due to an oversight of not updating the code with the latest version. Due to the fact this is a paper that focuses on the utility of the pipeline I cannot see it published without major correction to the code and revisions to the text.

Additional comments

I am happy to continue my review once the code has been fixed and I can run the pipeline end-to-end as instructed on the tutorial. As a bioinformaticians, we strive for both quality of the analysis and the reproducibility of the methods. I would like to see the author try and update both the text and the code.

·

Basic reporting

The manuscript is overall clearly written with effective visuals to facilitate interpretation.

Experimental design

Dr. Reginato presented a useful bioinformatic tool to harvest nuclear genes from genome skimming data. The concept of getting low copy nuclear genes from skimming data and the pipeline itself is not new to science, but the author made an effort to compile it into a user-friendly format. I think this will be an important contribution to the community, making multilocus phylogenetic analysis more accessible to researchers who are less experienced with bioinformatics. I appreciate the author’s effort to benchmark the parameter combinations and to explore the limits of the pipeline, but my bigger concern is the masking of polymorphic sites, which can be problematic when there are gene duplications. A few other issues also need to be addressed before it is ready for publication:

There are quite a few studies mining low copy nuclear genes from genome skimming than the author claimed in the introduction (Line 91-92). To name a few, Loiseau et al (2021) explored hybridization in Bromeliaceae; Liu et al (2021) reconstructed phylogeny of Vitaceae; Meng et al (2021) explored cyto-nuclear discordance in Rosaceae. The bioinformatic methods employed in Loiseau et al (2021) and Meng et al (2021) are very similar to skimmingLoci, and I think is even slightly better. I also want to mention that myself have developed a software to harvest low copy nuclear genes from genome skimming data with a different assembly algorithm (Cai et al, 2022; Github https://github.com/lmcai/PhyloHerb/#iv-retrieve-low-copy-nuclear-genes). These recent advances are not reflected in the current manuscript and need to be added. Given that this mapping–base calling method has been around for a few years, I thin the author need to give credit to researchers who pioneer this effort.

I am quite confused about the ‘Reference comparison’ section. The author claimed at the beginning to generate three reference sets. Their difference lies in their intron content. First, I would suggest replacing ‘transcripts’ with ‘CDS’ because ‘transcripts’ is an informal term, and it can refer to premature mRNA with introns. Second, I cannot find any descriptions on how these three types of references were generated and incorporated in the downstream comparison. In Line 192-194, only the ‘transcripts’ and ‘full genes’ were used instead of all three claimed at the beginning? I am curious how ‘full gene’ reference can be generated from the onekp data. I think all of these confusions suggest that major revision needs to be done to clearly describe the purpose and process of this reference generation step. A supplementary figure of the bioinformatic flowchart for this section can help readers to interpret the methods more easily. I also suggest the author to define ‘coverage’ at the very beginning because the meaning of coverage overlaps with ‘depth’. I was initially confused about the redundancy but later realize that coverage probably refers to sequence completeness instead.

My primary concern for the method is not the proportion of missing data, but the use of consensus sequences. In Loiseau et al (2021), efforts were made to phase the bases such that multiple copies of the same gene can be individually identified. However, skimmingLoci does not attempt to separate paralogous copies. What are the parameters used in bwa for reads mapping and those used in vcftools for variant calling? Will ambiguous sites be called? Using default values in these programs will be problematic when gene duplication is common. The author needs to either incorporate this phasing step in the pipeline, or demonstrate that polymorphism is neglectable. One way to benchmark this is to run the pipelines with different parameter settings in bwa. Then compare the proportion of polymorphic sites and species tree reconstructions, similar to the comparisons done with different minimum depths. Meanwhile, these caveats should be mentioned in the Discussion section. I think the readers would especially appreciate it when guidance is provided to use appropriate parameters to avoid paralogs.

Finally, the manuscript is overall clearly written but needs to be tightened up by removing redundancy. Some of them are pointed out in the minor comments below:

Minor comments:
Abstract: Make the name and link of the software available in the abstract. For people who just read the abstract, they should be able to easily find it online.
Line 18-19: Remove ‘early’ in Line 18, replace ‘genomic reduction step’ with ‘biased enrichment step’ in Line 19.
Line 23-24 The phrase ‘the trade-off between libraries with few reads and species with large genomes’ is confusing. Do you mean large amount of missing data? I think it might be more propriate to directly say ‘missing data caused by low coverage’.
Line 30 Remove ‘putative’?
Line 34 Remove ‘and depth’.
Line 35 Remove ‘reduces the amount of assembled data and’
Line 36 Move ‘in turn’ to the second clause.
Line 51 ‘Prior to’ instead of ‘prior’
Line 50-52 Reads somewhat awkward. Please revise the sentence.
Line 68 ‘A sample of genes’? Do you mean ‘a subset of genes’?
Line 167 The meaning of ‘-d -C -T’ was not described in the previous text. Either explain or remove.
Line 226 What is the difference between ‘missing data percent’ and ‘median coverage’?

Liming Cai


Reference
Cai, Liming, Hongrui Zhang, and Charles C. Davis. "PhyloHerb: A high‐throughput phylogenomic pipeline for processing genome skimming data." Applications in Plant Sciences (2022): e11475.

Liu, Bin‐Bin, et al. "Capturing single‐copy nuclear genes, organellar genomes, and nuclear ribosomal DNA from deep genome skimming data for plant phylogenetics: A case study in Vitaceae." Journal of Systematics and Evolution 59.5 (2021): 1124-1138.

Loiseau, Oriane, et al. "Genome skimming reveals widespread hybridization in a Neotropical flowering plant radiation." Frontiers in Ecology and Evolution 9 (2021): 668281.

Meng, Kai-Kai, et al. "Phylogenomic analyses based on genome-skimming data reveal cyto-nuclear discordance in the evolutionary history of Cotoneaster (Rosaceae)." Molecular Phylogenetics and Evolution 158 (2021): 107083.

Validity of the findings

Results support the claim.

---

## Round 0.2 · Minor Revisions

Dear Author

The reviewer has recommended revisions to your manuscript.

Therefore, I invite you to respond to the reviewer's comments and revise your manuscript. Especially the introduction.

With Thanks

·

Basic reporting

My pronouns are she/her/hers instead of he/him/his. The author could use more gender inclusive pronouns such as they/them/their when they are uncertain. I appreciate the effort that author has taken to revise the manuscript, including minor revisions for clarity in many cases. My comments are intended to improve the readability of the manuscript by encouraging the author to provide a comprehensive overview of the backgrounds and limitations of the method. Most of my concerns have been effectively addressed but there are still few important ones dismissed in the revised version. One of my major comments is that recent advances in harvesting nuclear loci from skimming data is not effective summarized in the introduction. The author needs to elaborate more on the similarities and distinctions compared to previous studies and how skimmingLoci allows for new avenues to be explored along these lines. I do not think that a method where one can cite twenty papers can be regarded to as ‘(an avenue) still largely neglected’. Secondly, the author clarified that ambiguous sites will be called IUPAC codes. As another byproduct of low coverage, ambiguous sites will be as prevalent as missing data, the letter of which has been extensively discussed in the manuscript. However, how ambiguity is handled by skimmingLoci was not discussed at all, neither was its potential impact on downstream inferences. I like the relative ambiguity histogram the author provided. It will be worthwhile to elaborate more on the potential caveats of skimmingLoci such as ambiguity and the best practice to mitigate these challenges. As a method and software developer, it is one’s duty to guide users through both highlights and pitfalls of the tool. It could be done relatively easy with one/two-sentence description in the methods and few more in the discussion. The references used to demonstrate the neglectable effect of phasing paralog can be added, along with those showing the opposite pattern (e.g., Andermann 2019 Sys Biol).

Experimental design

NA

Validity of the findings

NA

Additional comments

NA

---

## Round 0.3 · accepted · Accept

Dear Author

I am pleased to inform you that after the last round of revision, the manuscript has been improved a lot, and it can be accepted for publication.

Congratulations on the acceptance of your manuscript, and thank you for your interest in submitting your work to PeerJ

·

Basic reporting

I am satisfied with the revision. Congratulations to the author for bringing new tools to the field.

Experimental design

NA

Validity of the findings

NA

Additional comments

NA